# Invasive Traits of *Symphyotrichum squamatum* and *S. ciliatum*: Insights from Distribution Modeling, Reproductive Success, and Morpho-Structural Analysis

**DOI:** 10.3390/biology14010047

**Published:** 2025-01-09

**Authors:** Alina Georgiana Cîșlariu, Ciprian Claudiu Mânzu, Mioara Dumitrașcu, Daniela Clara Mihai, Marius Nicu Andronache, Petronela Camen-Comănescu, Eugenia Nagodă, Anca Sârbu

**Affiliations:** 1Faculty of Biology, University of Bucharest, 1-3 Intr. Portocalelor, 060101 Bucharest, Romania; alina.cislariu@unibuc.ro (A.G.C.); mioara.dumitrascu@bio.unibuc.ro (M.D.); daniela.m@bio.unibuc.ro (D.C.M.); m.andronache20@s.bio.unibuc.ro (M.N.A.); anchusa24@yahoo.com (A.S.); 2Faculty of Biology, “Alexandru Ioan Cuza” University of Iași, Bd. Carol I, No. 20 A, 700505 Iași, Romania; 3Faculty of Horticulture, University of Agronomic Sciences and Veterinary Medicine of Bucharest, 59 Mărăști Blvd., 011464 Bucharest, Romania; 4Botanical Garden “D. Brandza”, University of Bucharest, 32 Șos. Cotroceni, 060114 Bucharest, Romania; petronela.comanescu@bio.unibuc.ro (P.C.-C.); eugenia.nagoda@bio.unibuc.ro (E.N.)

**Keywords:** alien species, anatomy, ecology, environmental variable, habitat suitability modeling

## Abstract

Invasive plant species can disrupt ecosystems, reduce biodiversity, and even pose risks to human health. Understanding why certain plants become invasive is crucial for controlling their spread and minimizing their impact. Identifying a species’ potential to spread in the early stages of invasion is particularly important, as managing established invasive species is far more challenging, costly, and less effective. This study focuses on *Symphyotrichum squamatum,* a South American plant first recorded in Romania in 2015. It is highly invasive in other European countries, raising concerns about its possible invasion in Romania. In this study, we investigated its potential to become invasive by examining its distribution patterns in relation to bioclimatic factors, as well as its reproductive and anatomical traits. We also compared *S. squamatum* to *S. ciliatum*, a plant species already invasive in Romania, to understand shared traits that may contribute to invasiveness. Our findings revealed that while *S. squamatum* has similar traits with successful invaders, Romania’s climate may limit its further spread. This research enhances our ability to predict and manage plant invasions, which is essential for protecting ecosystems and reducing the impact of invasive species.

## 1. Introduction

Biological invasions are a major driver of biodiversity loss and ecosystem degradation worldwide [1]. Invasive alien species (IAS) disrupt ecosystem functions by altering native community composition, biotic interactions, and biodiversity, often transforming habitat structure and function [2,3,4]. These disruptions occur through various mechanisms, including resource competition [5,6,7], excessive resource consumption [8], allelopathy [9,10], and interference in mutualistic relationships [11,12]. Human activities, including agriculture, aquaculture, and transportation, facilitate the introduction and spread of non-indigenous species across natural dispersal barriers. While many species fail to establish in new environments, those that succeed can threaten human health and cause significant economic damage [13].

In Europe, the issue of IAS gained visibility in the late 1990s following environmental audits that highlighted their extensive impacts [14,15,16]. By 1998, the European Community Biodiversity Strategy recognized IAS as a critical environmental issue, and in 2002, the European Council identified them as a leading cause of biodiversity loss and economic damage [17,18]. Efforts to address IAS have since included the 2002 European Strategy on IAS under the Bern Convention and legislative measures within the EU Habitats Directive [17,19]. Despite these efforts, challenges such as inconsistent monitoring, fragmented data, and difficulties in standardizing species classifications remain significant, highlighting the necessity for centralized coordination to address biological invasions in the European Union [20].

Two IAS that exemplify these challenges are *Symphyotrichum squamatum* (Spreng) G.L.Nesom and *Symphyotrichum ciliatum* (Ledeb.) G.L.Nesom, species posing significant ecological risks. *Symphyotrichum squamatum*, native to South America [21], is known for its rapid spread and aggressive behavior, forming dense populations that outcompete native flora and disrupt local biodiversity [22,23,24,25]. Similarly, *S. ciliatum*, originating from North America, thrives in moist, saline environments, where it forms mono-dominant stands that outcompete native plants and potentially alter habitat composition [26]. Both species are currently widespread in Europe, yet their invasive dynamics in specific regions, including Romania, require further investigation.

Understanding the traits that enhance a species’ invasive potential represents the key to predicting and managing biological invasions [27]. Early research focused on biological characters associated with successful IAS establishment, including fitness homeostasis [28], extended flowering and fruiting periods [29], and high propagule pressure [30,31,32,33]. Vegetative reproduction, often a key driver of plant invasions, has been widely documented as a contributing factor [34,35,36,37]. Additionally, a species’ large native range is considered another indicator of potential invasiveness [38]. As predictive tools for assessing invasion risk evolved, research began to emphasize the integration of quantitative methods, ecological models, and early-stage monitoring to identify the probable invaders and areas at risk [13,27,39,40,41]. Invasive species distribution models (iSDMS) have since emerged as powerful tools for managing IAS [42,43,44,45,46]. Recent studies have also highlighted specific structural features of alien species that may contribute to their invasiveness. For instance, a high Relative Transpiration Area (RTA) index has been proposed as a predictor for the transition of alien species into invasive species [47]. Traits such as robust stem structure and well-developed belowground organs are linked to adaptability and rapid evolution in heterogeneous environments [48]. Despite these insights, predicting invasion success remains complex due to the multitude of factors influencing each stage of the invasion process. Consequently, ongoing research emphasizes the need for early intervention and effective management strategies to mitigate the long-term ecological and economic impacts of IAS [13,49].

In light of these challenges, our study provides a holistic framework for assessing the invasion potential of *S. squamatum* in Romania by integrating multiple methodologies. Using a comparative framework, we combine species distribution modeling (SDM), a reproductive trait analysis, and anatomical studies to comprehensively assess a species’ invasive ability. By comparing *S. squamatum*, a recently introduced species, with *S. ciliatum*, a species already invasive in Romania, we aim to identify shared and distinct traits that influence their success.

This multifaceted approach links trait-based mechanisms of invasiveness with spatial predictions, which are often treated separately in the current literature [1,39,40,41,48,50]. We believe that this comparative analysis will allow us to pinpoint traits that may signal invasive potential, even before a species becomes widespread.

Our objectives are as follows: (1) to assess habitat suitability for both species in Romania using bioclimatic conditions and species distribution modeling (SDM); (2) to compare the structural and reproductive traits of *S. squamatum* and *S. ciliatum* and identify those associated with their invasive potential; (3) to integrate these findings into a comprehensive framework for predicting the invasive potential of *S. squamatum* in Romania.

By combining ecological modeling with a trait-based analysis, our study offers a robust framework for assessing the invasiveness of alien species, which can inform management strategies.

## 2. Materials and Methods

### 2.1. Species Description

*Symphyotrichum squamatum* belongs to the Asteraceae (Compositae) family [51]. Originally native to South and Central America [21,52], *S. squamatum* was first reported in Europe in Belgium in 1895, likely introduced for ornamental purposes [52,53,54]. Currently, it is widely naturalized in Europe, particularly in the South-West and Mediterranean regions [55]. It is listed as invasive in several European countries, including Greece, Italy, Spain, Portugal, France, Malta, Slovenia, Montenegro, Bulgaria, and Bosnia and Herzegovina [55]. In Romania, *S. squamatum* was first reported in 2016 near the Botanic Garden “D. Brandza” in Bucharest, though it is not part of the garden’s collections [56]. Currently, it is found in a few locations across Bucharest and Giurgiu County [57] (Figure 1).

*Symphyotrichum squamatum* is characterized as an annual or perennial plant with an erect stem, reaching up to 2 m in height. Inflorescences contain 50 to 500 florets per capitulum and the achenes have a non-accrescent pappus [21,25].

The species thrives in tropical, subtropical, and Mediterranean regions [52]. Its native habitats include moist to wet, saline areas, near beaches, ballast dumps [21], and salt marshes [58]. The species exhibits high salt tolerance and has been reported on a variety of humid soils, including loam, peaty sand, deep white sand, granite, laterite, and basalt soil [59,60,61]. It also demonstrates a great resilience to environmental conditions and is commonly found in semi-natural and disturbed habitats, particularly in freshwater, agricultural, and ruderal areas; along streets, sidewalks, and roadsides; and in industrial areas, flowerbeds, and weather banks [21,22,58,60].

Native to North America, *S. ciliatum* is considered to have migrated from the Rocky Mountains in Alaska and reached Northeast Asia [26,62,63,64]. According to Greuter [65], *S. ciliatum* is also considered native to Ukraine and European Russia [64]. However, according to the Plant of the World Online (POWO) database [51], the species is classified as introduced in East European Russia, Poland, Romania, and Ukraine, with its native range extending to Central America and Asia.

In Romania, *S. ciliatum* was first reported in 1967 in the eastern part of the country [66]. Since then, it has spread to other regions [64,67] (Figure 2) and is currently considered as an invasive species [66,67,68].

*Symphyotrichum ciliatum* is an annual plant [67,70] that can reach heights up to 5–70 (100) cm [62,67]. It features a taproot; its stem is erect and more or less succulent [62,67,70]. It thrives in light but can tolerate shade for short periods. The species can grow on brackish soil, and in wetlands, prairies, steppe areas, and even salt marshes, while preferred soils range from neutral to weakly acidic or weakly alkaline, or mesohaline, with a low to medium content of chloride salts [62,71]. Additionally, it has been recorded in irrigation canals; along railroad tracks, roadsides, and wastelands; and in brush ponds that dry up in the summer [62], construction sites, and quarries [63,70] or on ruderal lands near dams [72].

### 2.2. Ensemble Species Distribution Modeling (ESDM)

To assess the potential distribution of *S. squamatum* and *S. ciliatum*, we employed Ensemble Species Distribution Modeling (ESDM) using the sdm package in R [73], using the following modeling algorithms: the Generalized Additive Model (GAM), Multivariate Adaptive Regression Splines (MARSs), Random Forest (RF), and Boosted Regression Trees (BRTs). The models were calibrated using occurrence data from the species’ native and invaded ranges since research indicates that models trained on data from both native and invaded ranges yield more accurate predictions of a species’ spread compared to those based solely on native range data [74,75]. For the ESDM, we used 19 bioclimatic variables downloaded from WorldClim at a resolution of 2.5 arcminutes [76,77]. Prior to running the ESDM, we used the Variance Inflation Factor (VIF) function in R to check for correlations among 19 predictor variables, excluding those with VIF values exceeding 0.75. The final bioclimatic set included 11 variables for *S. squamatum* and 10 variables for *S. ciliatum* (Appendix A).

Occurrence data for the species were sourced from the Global Biodiversity Information Facility (GBIF) [69]. Additionally, we incorporated field records of *S. squamatum* collected during our field study in Romania (2014–2024). The dataset was refined through several steps, including filtering for reliable records from human observations and preserved specimens. We then applied spatial thinning using the spThin package [78], maintaining a minimum distance of 10 km between occurrences to reduce spatial autocorrelation. The data were further cleaned by identifying and removing duplicates and handling missing values. The CoordinateCleaner package [79] was employed to eliminate potentially erroneous georeferenced records, such as those located at country centroids, in the sea, or flagged by GBIF as problematic. Through this cleaning process, 246 records were flagged and removed, resulting in a final dataset of 2078 occurrences for *S. squamatum* and 1089 occurrences for *S. ciliatum*.

To account for the geographic and environmental disparities between the native and invaded ranges of *S. squamatum*, we generated pseudo-absences (PAs) using tailored strategies for each region. In the native range, with 303 occurrence points, we randomly generated 3000 PAs within the environmental space, maintaining a 10:1 ratio of pseudo-absence to presence points. For Europe, with 1393 occurrence points and more complex environmental heterogeneity, we randomly generated 10,000 PAs, following established guidelines for species distribution modeling [80,81]. To refine the pseudo-absence selection, we applied a filtering step to ensure that all PAs were located at least 5 km away from known occurrence points, avoiding overlap with presence data [81,82]. A similar approach was used for *S. ciliatum*, generating 10,000 PAs for its native range (for 751 occurrences) and 2000 PAs for Europe (for 193 occurrences).

Each modeling algorithm underwent ten repetitions, with models trained on 75% of the data and evaluated on the remaining 25% [83,84]. The model ensembles trained on the native range of the species were afterwards projected onto environmental conditions specific to Europe, and binary presence/absence maps were created using the TSS (True Skill Statistic) as the threshold.

Model performance was evaluated using several metrics, including the Area Under the Curve (AUC), correlation (COR), deviance, True Skill Statistic (TSS), sensitivity, and specificity. Models with an AUC > 0.85 were considered the best performers, while those below this threshold were excluded.

Variable importance was assessed using the getVarImp function from the sdm package, which calculates the relative importance based on the ensemble predictions of all models [73]. Response curves for the key environmental variables were generated using the rcurve function from the sdm package, visualizing the predicted probability of occurrence in relation to the values of the environmental predictors [73].

### 2.3. Environmental Context of Populations from Romania

To assess the environmental context of *Symphyotrichum squamatum* and *S. ciliatum* in Romania, we selected a range of environmental variables related to their distribution, including climatic-, pedologic-, human impact-, and topographic-related predictors.

For the bioclimatic analysis, we focused on the key bioclimatic variables that contributed the most to the Ensemble Species Distribution Models (ESDMs) for each species.

The Human Impact Index (HII) was sourced from the Center for International Earth Science Information Network (CIESIN)—Wildlife Conservation Society (Last of the Wild Data v2-2005 LTW2 Global Human Footprint Data set), at a resolution of 1 km^2^ (Wildlife Conservation Society—WCS) [85].

Additionally, we compiled several vector datasets for creating essential spatial variables, including rivers, land use classes, soil classes, and soil textures, sourced from Open Street Map Contributors (OSM) at a 1:1000 scale; Corine Land Cover 2018 [86] at a 1:100,000 scale; and the European Soil Data Centre (ESDAC) (Atlas of Romania Soils Map, 1:1,000,000) [87]. The Corine Land Cover (CLC) data were reclassified to better reflect the land use categories occupied by the species. Rivers were used to generate distance variables, adjusted for elevation variations with ArcGIS v.10.4.

### 2.4. Reproductive Trait Assessment of S. squamatum

Assessing the reproductive success of *S. squamatum* involved conducting germination tests on seeds collected from two populations in Romania: the Cotroceni and Rahova populations (Bucharest). These populations were selected based on the observed trends over several years. Notably, the Cotroceni population exhibited significant growth from just a few individuals to over 100 within a couple of years. Conversely, the Rahova population remained relatively stable, with approximately ten individuals, despite the available space for expansion in its wasteland habitat (similar to the Cotroceni population). This led us to consider potential factors affecting the reproductive success of the constant Rahova population over time.

For the germination experiment, a total of 800 seeds (400 seeds from each population) were selected. From each population, we planted 100 fully matured and healthy-looking seeds on soil, exposed to a controlled temperature of 21 °C during the day and 8 °C at night, with a 16/8 h photoperiod (Experiment 1), which are consistent with International Seed Testing Association (ISTA) standards for seed germination for the species of the *Symphyotrichum* genus [88] and also align with the environmental parameters typical for Romania’s growing season, thus simulating both optimal laboratory standards and relevant natural conditions for germination in the invaded range. Another 100 seeds were planted on soil exposed to a constant room temperature of 23 °C (Experiment 2). Additionally, 100 seeds from each population were placed on Petri dishes lined with moist filter paper, also subjected to the same temperature conditions.

Prior to the experiment setup, the seeds underwent sterilization to prevent potential contamination, with commercial bleach (active Cl concentrations less than 5%) for 5 min, followed by thorough rinsing with sterile distilled water (4 times). The soil used was autoclaved to eliminate any potential infestations, as were the Petri dishes along with the filter papers. Throughout the 20-day duration of the experiment, regular watering with sterile distilled water was maintained, and daily observations were recorded to monitor progress.

Reproductive success was evaluated by calculating the following germination parameters for both Cotroceni and Rahova populations:

The Germination Percentage (GP), which reflects seed viability by indicating the proportion of seeds that successfully germinate, was calculated as [89]:$$\mathrm{GP}=\frac{Number\;of\;germinated\;seeds}{Total\;number\;of\;seeds\;planted}\times 100.$$

Germination Energy (GE), which estimates seed vigor during the early stages of germination, was calculated as [89]:$$\mathrm{GE}=\frac{Number\;of\;seeds\;germinated\;on\;the\;4th\;day}{Total\;number\;of\;seeds\;planted}\times 100.$$

The Germination Rate Index (GRI), which reflects both the speed and uniformity of germination, with higher values indicating faster and higher germination, was calculated as$$\mathrm{GRI}=\frac{G1}{1}+\frac{G2}{2}+\ldots +\frac{Gx}{x}$$
where G1, G2, …, Gx represent the percentage of seeds germinated on the first, second, …, and x^th^ day after sowing [90].

### 2.5. Comparative Analysis of Anatomical Traits

For the comparative analysis of the anatomical traits of *S. squamatum* and *S. ciliatum*, we relied on the investigations of Sârbu and Smarandache (2015) [67] regarding the vegetative body anatomy of *S. ciliatum* plants collected from Sacalin Island (Sfântu Gheorghe, Danube Delta, Tulcea County, Romania), and Dumitraşcu et al. (2023) [91] on *S. squamatum* plants collected from Bucharest (Şos. Cotroceni, 32), Romania.

The biological material from both species underwent similar preservation methods (70% alcohol), manual cross-sectioning (at standardized levels: median third of the primary root, lower and upper thirds of the main stem, and median third of the lamina), and staining (using Iodine Green and Carmin Alum, following the double-staining technique described by Şerbănescu-Jitariu et al. (1983) [92]), and were subsequently micro-photographed with a DOCUVAL (Carl Zeiss, Jena, Germany) optical microscope with an incorporated NIKON D90 digital camera.. Additionally, we analyzed supplementary cross-section photographs (besides those published) for assessing the measurements of the analyzed parameters of the vegetative organs.

## 3. Results

### 3.1. Ensemble Species Distribution Modeling

The Ensemble Species Distribution Models (ESDMs) developed using occurrence data from both the native and invaded range of *S. squamatum* and *S. ciliatum*, along with bioclimatic variables, demonstrated excellent performance (Table 1).

The evaluation metrics indicate strong model accuracy and reliability. Specifically, the AUC (Area Under the Curve) values of both species (Table 1) reflect an excellent fit of the models to the observed data, suggesting high predictive power [93]. Additionally, other metrics such as correlation, deviance, and the True Skill Statistic (TSS) further confirm the reliability of the models in predicting the species’ potential distributions.

The ESDM identified extensive areas in Europe as climatically suitable for *S. squamatum*, particularly across countries such as Italy, Spain, Portugal, France, Croatia, Montenegro, Albania, Greece, Bulgaria, Turkey, and Bosnia and Herzegovina (Figure 3). Notably, many of these countries, including Greece, Italy, Spain, Portugal, France, Montenegro, Bulgaria, and Bosnia and Herzegovina, have already reported *S. squamatum* as an invasive species [94,95], further supporting the accuracy of the model. The identified suitable areas are predominantly characterized by temperate climates, with dry summers, under Mediterranean or oceanic influences, or by temperate, fully humid climates. Conversely, areas with continental temperate climates, such as inland or high-altitude regions, are excluded from favorable zones (Figure 3). Based on continentality, these areas are euoceanic, semihyperoceanic, or semicontinental [96,97,98].

In Romania, the model predicts very limited climatically suitable areas for *S. squamatum*, primarily located in the southern regions around Bucharest and Giurgiu, where the species has been recorded in the field (Figure 1).

In its native range, the model identifies suitable conditions for *S. squamatum* in southeastern South America, particularly in temperate regions of Argentina, Uruguay, and southern Brazil (Figure A1). These areas are characterized by temperate climates, fully humid, with consistent annual rainfall and oceanic influences in coastal areas. Based on the continentality types, these zones are classified as euoceanic, semihyperoceanic, or semicontinental [96,97,98].

A variable contribution analysis showed that Bio 1 (Annual Mean Temperature) had the highest contribution to the ESDM of *S. squamatum*, followed by Bio 3 (Isothermality) (Appendix A). Other influential variables included Bio 5 (Max Temperature of Warmest Month), Bio 9 (Mean Temperature of Driest Quarter), and Bio 2 (Mean Diurnal Range) (Appendix A). These results indicate that temperature-related variables are the key drivers of the species’ modeled potential distribution.

The response curve of Bio 1 (Annual Mean Temperature) shows optimal habitat suitability around 15 °C (Figure 4), with the species tolerating higher temperatures, but being less suited to extreme heat. The response curves of the other influential variables highlight that the species prefers regions with moderate temperature stability, tolerates high temperatures during the warmest periods, is favored by moderately dry conditions, and is associated with environments with moderate day-to-night temperature variations (Figure 4).

Climatically suitable areas for *S. ciliatum* in Europe, as identified by the ESDM, include regions in Romania, Hungary, Ukraine, Bulgaria, the Republic of Moldova, European Russia, and Poland (Figure 5). These areas are characterized mostly by a continental temperate climate, cooler and fully humid. The continentality types are subcontinental and eucontinental [96,97,98].

In its native range, the model identifies suitable conditions in the central United States, southern Canada, parts of the Rocky Mountain regions, and Central Asia (Figure A2). These areas are generally characterized by continental temperate to semi-arid climates with wide temperature ranges, moderate precipitation, and occasional semi-arid conditions [96,97,98]. The North American climate appears to be more similar to the invaded European one (mostly cold, fully humid), while Central Asia features a cold climate, with dry winter. Across the native, naturalized, and invaded ranges, the areas share similar continentality types—eucontinental and subcontinental [96,97,98]. Unlike *S. squamatum*, *S. ciliatum* shows more extensive bioclimatic suitability across Romania (Figure 5).

The variable contribution analysis showed that Bio 1 (Annual Mean Temperature) was the most significant factor influencing the ESDM for *S. ciliatum*, followed by Bio 5 (Max Temperature of Warmest Month) (Appendix A). Other contributing variables included Bio 4 (Temperature Seasonality), Bio 2 (Mean Diurnal Range), and Bio 3 (Isothermality) (Appendix A). As with *S. squamatum*, temperature-related variables are the primary drivers of the species’ modeled potential distribution.

The response curves for key environmental variables derived from the ESDM (Figure 6) highlight *S. ciliatum*’s preference for moderate to cool annual temperatures. The model also suggests that the species can tolerate high temperatures during the warmest periods, and favors environments with moderate day-to-night temperature fluctuations, as well as moderate levels of Isothermality.

### 3.2. Environmental Context of Populations from Romania

We compared the key bioclimatic factors associated with *S. squamatum* and *S. ciliatum* occurrences in Romania to those in their native and invaded ranges (Table 2 and Table 3), using average and standard deviation values for the most influential ESDM variables. For *S. ciliatum*, the native range includes occurrences from Central America and Asia [51].

The Annual Mean Temperature (Bio 1) that corresponds to the occurrences of *S. squamatum* in Romania is much lower than in its native range and in the European invaded range (Table 2). Comparing these values with the response curve for Bio 1 (Figure 4), at 10 °C, habitat suitability for the species is very low. These colder Annual Mean Temperatures most likely represent a limiting factor for the species’ establishment and spreading in Romania.

Additionally, Isothermality in Romania is substantially lower than in both the native and invaded ranges, which indicates that the variability in day-to-night and seasonal temperature fluctuations in Romania’s climate may further inhibit the species’ ability to thrive, being a possible contributor to the absence of extensive suitable bioclimatic areas for this species.

The mean temperature during the driest quarter (Bio 9) is also much lower in Romania compared to the native and invaded ranges. This cold stress during a critical dry period could adversely affect the species’ growth and survival.

The average values for the Mean Diurnal Range (Bio 2) and Max Temperature of the Warmest Month (Bio 5) for *S. squamatum* in Romania are comparable to those observed in both its native and invaded ranges (Table 2). These values also align with the range associated with habitat suitability, as indicated by the response curves for these bioclimatic variables (Figure 4).

In contrast, the bioclimatic conditions for *S. ciliatum* in Romania are more favorable, aligning with those found in both its native and European invaded ranges (Table 3). These conditions also correspond to the values identified as suitable in the response curves for the bioclimatic variables that contributed the most to the ESDM (Bio 1, Bio 5, Bio 4) (Figure 6).

To provide a clearer overview of the environmental conditions that characterize both species in Romania, we further analyzed soil characteristics, human impact, and topographic factors associated with their populations.

Analyses of soil classes and textures associated with species occurrences revealed that *S. squamatum* inhabits disturbed or anthropogenically influenced soils in urban areas. In contrast, *S. ciliatum* primarily grows on Mollisols (Appendix A) but also inhabits a variety of disturbed or altered soils. *Symphyotrichum squamatum* is commonly found in clayey-silt and sandy-loam soils, while *S. ciliatum* occupies clay-rich and heterogeneous soil texture. The wider distribution of *S. ciliatum* across various soil classes and textures (Appendix A) suggests greater adaptability to diverse soil conditions compared to *S. squamatum* in Romania.

Both the Human Impact Index (HII) and land use analysis indicate that *S. squamatum* mainly occurs in artificial areas such as settlements and near infrastructure (Appendix A), with moderate to moderate–high levels of anthropogenic impact, whereas *S. ciliatum* is more widely distributed across diverse land use types, including arable land, forests, shrublands, vineyards, and disturbed areas like dump sites. This suggests that *S. ciliatum* has greater adaptability to a variety of human-altered landscapes.

### 3.3. Reproductive Success of S. squamatum

The results of the germination experiment revealed differences in germination characteristics between the two *S. squamatum* populations (Rahova and Cotroceni) and across substrate types and temperature treatments (Table 4).

The highest Germination Percentage (GP) of 73% was recorded for Rahova seeds planted in soil under controlled temperature conditions (Experiment 1) (Table 4). Conversely, the lowest GP of 12% was observed in the Cotroceni population, where seeds were placed on Petri dishes with moist filter paper under room temperature conditions (Experiment 2) (Table 4).

Germination dynamics showed that seeds generally began germinating on the 3rd day of the experiment, with germination ceasing around the 14th to 15th day (Figure 7). However, seeds from Rahova exposed to constant room temperature (Experiment 2) exhibited delayed germination compared to other treatments.

In terms of Germination Energy (GE), seeds from the Cotroceni population exhibited higher initial vigor across all treatments, with notably higher GE compared to those from Rahova (Table 4). This pattern was also reflected in the Germination Rate Index (GRI), where Cotroceni seeds demonstrated faster and more consistent germination rates compared to Rahova seeds, except for Cotroceni Soil Experiment 2, where GRI was markedly lower (Table 4).

### 3.4. Comparative Analysis of Anatomical Traits

Both *S. squamatum* and *S. ciliatum* exhibit a secondary structure in their roots and feature a cortical aeriferous tissue. The aeriferous tissue is better developed and organized in *S. squamatum* compared to *S. ciliatum*. Thus, *S. ciliatum* features a parenchymatous cortex with large cells, meatuses, and aeriferous gaps, constituting an aerenchyma covering approximately 31.7% of the root surface (Appendix A). *Symphyotrichum squamatum* displays a cortical aerenchyma with aeriferous channels, covering about 58.2% of the root surface (Appendix A). In contrast, *S. ciliatum* exhibits a better developed xylem, covering approximately 37.3% of the root surface, while *S. squamatum* roots’ xylem covers only about 11% (Appendix A).

Both species exhibit a secondary structure in their stems, though with species-specific differences. In *S. ciliatum*, the secondary structure of the stem is represented by an annual circular ring at both the base and top. Conversely, *S. squamatum* presents a circular annual ring at the stem base and a fascicular secondary structure at the top.

Throughout the stem length, *S. ciliatum* consistently displays a better-developed xylem compared to *S. squamatum*, which shows a reduction in xylem coverage from the base to top. Aerenchyma is present in the stem of both species, with *S. squamatum* exhibiting a more uniform and developed distribution along the entire stem length. In contrast, *S. ciliatum* shows a decreasing coverage of aerenchyma from the base to top of the stem. Additionally, medullary aeriferous cavities are present in the median and upper stem areas of *S. ciliatum*, a feature absent in *S. squamatum* [67,91].

The leaf lamina of both species is bifacial, and amphystomatic, and displays an equifacial structure. The vascular bundles are surrounded by parenchymatous non-chlorophyllous sheaths (acquiferous structures). In *S. ciliatum*, the median vascular bundle connects to both epidermis through parenchymatous cell columns, whereas in *S. squamatum*, the median bundle features a well-developed collenchyma. The median conducting bundle in *S. squamatum* is larger compared to *S. ciliatum* (Appendix A). Mesophyll characteristics are similar between the two species (Appendix A). Stomata in *S. ciliatum* are positioned at approximately the same level as the epidermal cells, whereas in *S. squamatum*, stomata are situated at a higher level [67,91].

## 4. Discussion

### 4.1. Ensemble Species Distribution Modeling

The results of the Ensemble Species Distribution Model (ESDM) underscore the significant invasive potential of *S. squamatum* mainly in southern Europe, with extensive areas identified as potentially suitable habitats across multiple countries (Figure 1). The species shows a preference for moderately warm, stable temperature conditions, consistent with its known invaded area [52,55,94,95].

In Romania, however, the model indicates that suitable climatic conditions for *S. squamatum* are very limited, concentrated in the southern regions near urban centers, where its presence has been documented [56,57]. Additionally, the cooler annual temperatures, significant diurnal fluctuations, and cold, dry conditions that feature Romanian climate [99] may limit the species’ growth and potential for further dispersal.

In contrast, *S. ciliatum* appears to be well suited to regions with pronounced temperature variations across seasons, consistent with regions that experience distinct summers and winters. These results highlight *S. ciliatum*’s adaptability to diverse climatic conditions, particularly in regions with temperate and continental climates. The combination of moderate temperatures and seasonal variability within its tolerance range suggests that Romania provides suitable habitats for the species, indicating its potential for establishment and further spread.

Both species’ ESDMs suggest that continentality plays a key role in their distribution. For *S. ciliatum*, reduced continentality and increased oceanic influence seem to limit its expansion to western Europe. Conversely, an increase in continentality appears to constrain *S. squamatum*’s spread to central and northern Europe.

It is important to note that the potential invasiveness of *S. squamatum* does not depend solely on climatic factors. Instead, its spread is likely influenced by a complex interaction of factors, including habitat suitability, dispersal mechanisms, and anthropogenic influences [100,101].

### 4.2. Environmental Context of Populations from Romania

Trait-based approaches to studying invasion success often compare the traits of invasive species with those of native or non-invasive alien species [102]. These comparisons are particularly effective for identifying characteristics that predispose species to invasiveness, with studies suggesting that invasive to non-invasive comparisons reveal more predictive traits than invasive–native comparisons [103,104].

Our results show that both *S. ciliatum* and *S. squamatum* occupy disturbed or anthropogenically altered habitats in Romania, but they differ in their adaptability. *S. ciliatum* exhibits a broader environmental tolerance, occurring across various land use types, adapting to diverse soil conditions, and tolerating a wider range of anthropogenic pressures. It also demonstrates greater flexibility in its proximity to watercourses, enabling it to occupy a wider variety of habitats. Invasive plants are often characterized by their ability to thrive in diverse environmental conditions, such as variations in the substrate type, moisture levels, light availability, and temperature regimes [105]. This broad ecological tolerance has allowed *S. ciliatum* to establish itself in a wide range of environments, as reflected in its extensive distribution in Romania. These findings suggest that *S. ciliatum* is more likely to continue expanding its range in Romania, while the future spread of *S. squamatum* remains uncertain.

In contrast, *S. squamatum* is primarily found in environments with moderate to moderate–high anthropogenic impact, particularly in urban areas and infrastructure zones. This suggests that while *S. squamatum* is able to occupy niches where other species may struggle, its spread in Romania is likely constrained by specific environmental requirements. The species’ presence in only two regions (Giurgiu County and Bucharest) may be due to their strategic location as access points between southern Europe and Romania, as the species has been reported as naturalized in Bulgaria and Turkey.

Research on alien species consistently demonstrates a pattern: the longer alien species inhabit their introduced ranges, the higher the probability of their widespread establishment [106,107]. This temporal dimension might also explain the current invasive status of *S. ciliatum*, which was introduced in Romania around 1967, compared to *S. squamatum*, which was first reported in 2016. The longer period since introduction has likely allowed *S. ciliatum* to spread across various regions, contributing to its invasive success.

### 4.3. Reproductive Success of S. squamatum

The invasion success of alien species is attributed to several factors such as non-specialized germination requirements; rapid growth; high seed production, some of which may remain viable in soil seed banks for extended periods [105]; a prolonged period of flowering [108,109]; and minimal dependence on pollinators [104,110]. The establishment of naturalized populations depends on environmental compatibility between the introduced species and the new habitat, including climatic niche overlap and the ability to meet reproductive needs. Successful invaders typically exhibit traits that enhance dispersibility and germination efficiency, contributing to their dominance in new environments [111].

Germination tests conducted on *S. squamatum* seeds from two distinct Romanian populations, Cotroceni (with over 100 individuals) and Rahova (with just around 10 individuals), demonstrated the species’ ability to germinate under diverse conditions. While seeds from the Rahova population showed higher Germination Percentages (GPs) across most treatments, Cotroceni seeds generally exhibited a superior Germination Energy (GE) and Germination Rate Index (GRI), indicating greater vigor and faster germination. These results highlight the species’ non-specialized germination requirements, with both populations achieving significant germination rates across varied substrates and temperature conditions, irrespective of population size(Table 4).

*Symphyotrichum squamatum*’s reproductive traits further underscore its invasive potential through a prolonged flowering period from June to October, with fruiting extending until November and a significant seed production (up to 70,000 seeds per mature plant), facilitating extensive seed dispersal through wind [58]. Despite the absence of comparable germination studies from other regions, observations of Tripathi and Sharma (2019) [25] of numerous plantlets around parent plants in the subsequent season confirm the species’ high germination rate.

The comparable literature on *S. ciliatum*’s germination capacity is lacking. However, its reproductive traits, including a prolonged flowering period from August to October [112,113], autogamous fertilization [70], and wind-dispersed achenes [114], along with the ability of seeds to remain viable for up to ten years [70], indicate a probable similar capacity for wide dispersal and successful germination. The timing of flowering and fruiting, coinciding with periods of strong winds, may enhance *S. ciliatum*’s invasive potential through efficient anemochorous seed dispersal [115,116].

Considering these traits, it appears that both *S. squamatum* and *S. ciliatum* possess characteristics for their widespread establishment. With their prolific seed production, these species exhibit features typical of successful invaders.

### 4.4. Comparative Analysis of Anatomical Traits

Both species exhibited structural adaptations to the humid, wet, and saline environments [67,91], which align with their known ecological preferences and requirements from their native ranges: *S. squamatum* thrives in moist to wet saline areas and salt marshes [21,58], while *S. ciliatum* is found in wetlands within prairies, steppe areas, and salt marshes, and along irrigation channels [62].

The anatomical features of *S. squamatum* and *S. ciliatum* showed distinct adaptations, reflecting their ecological preferences and competitive strategies.

Notably, both species exhibit a well-developed secondary structure in their roots and stems, despite the fact that they both are annual species. This contributes to their robustness and rapid growth, providing the species with competitive advantages [48]. This robust stem architecture supports the weight of inflorescences predominantly found on the upper parts of the stem, ensuring efficient nutrient transport, a trait observed in other invasive species such as *Solidago canadensis* L. [48]. According to Gioria and Osborne (2014) [117], traits associated with faster growth and rapid resource acquisition appear to promote invasiveness.

A notable structural difference between *S. squamatum* and *S. ciliatum* lies in the development of aerenchyma within their root systems, which may reflect different strategies for oxygen transport in varying soil moisture and waterlogged conditions [48]. The significant cortical aerenchyma in *S. squamatum* suggests an adaptation that enhances its ability to tolerate waterlogged environments, which may provide a competitive advantage in habitats with fluctuating water levels or poor oxygen availability in the soil. In contrast, *S. ciliatum* exhibits smaller but still substantial aerenchyma coverage, likely an adaptation to tolerate waterlogged soils while maintaining root structural integrity [48].

Another notable observation is the contrasting xylem development between the two species. Throughout the stem length, *S. ciliatum* consistently displays a well-developed xylem compared to *S. squamatum*, which exhibits a reduction in xylem coverage from the base to top [67]. This divergence in xylem distribution could indicate differing strategies for water transport efficiency and structural support, essential for plants thriving in various soil conditions and wind exposure [48].

Moreover, the presence of sclerenchymatous tissues in the stem of invasive species serves as an indicator of their invasiveness [118]. These tissues, forming caps around vascular tissues, contribute significantly to the resilience of invasive species against cavitation, embolism, and implosion [119]. In the stem of both *S. squamatum* and *S. ciliatum*, the vascular tissues form caps of sclerenchymatous fibers [67,91].

The anatomical traits of *S. squamatum* and *S. ciliatum* reflect their competitive strategies, with features such as robust secondary structure, well-developed aerenchyma, and sclerenchyma tissues, enhancing their adaptability to diverse habitats. These traits not only provide ecological advantages but also align with characteristics observed in successful invaders, highlighting their potential resilience.

## 5. Conclusions

The integration of species distribution modeling (SDM), a reproductive analysis, and anatomical studies provides a robust framework for assessing species invasiveness. Species distribution modeling (SDM) helps identify potential risk areas based on bioclimatic compatibility, while an anatomical analysis reveals key structural adaptations that confer competitive advantages in new environments. Reproductive biology, particularly germination tests, is critical for understanding a species’ ability to establish and persist across diverse environmental conditions. By combining these approaches, we bridge the gap between morpho-anatomy and distribution data, creating a holistic framework for assessing IAS invasiveness. These integrated methods enhance our ability to predict invasion success and guide management strategies.

Suitability versus survivability: Our findings showed that while *S. squamatum* shares many characteristics with successful invaders, as seen in its comparison with the invasive *S. ciliatum*, its expansion in Romania may be constrained by environmental limitations. Both species exhibit characteristics of successful invaders (annual life cycle, extended flowering period, high fruit production, wind dispersal, high germination rate, and ecologically advantageous anatomical structures), but only *S. ciliatum* benefits from bioclimatic-favorable conditions. Continentality emerges as a key factor shaping the distribution of both species, though with differing effects. *S. ciliatum* appears to be better adapted to the broader continental characteristics of the Romanian climate, whereas *S. squamatum* is restricted by its preference for more stable and moderate conditions. At a broader scale, reduced continentality and increased oceanic influence limit the expansion of *S. ciliatum* into western Europe, while greater continentality constrains *S. squamatum*’s spread into central and northern Europe. The potential climatic niche and suitable habitats suggest that *S. ciliatum* will continue to expand in regions of Romania with a continental climate. In contrast, *S. squamatum* can currently be regarded as an opportunistic alien species in Romania, surviving where it is established, due to its resilience based on intrinsic anatomical and biological features.

Future research should focus on exploring the long-term ecological impacts of *S. squamatum* and its adaptability to Romania’s climate, while continuing to enforce control measures for *S. ciliatum* in regions where it has already been established. Identifying hotspots for the introduction and concentration of alien species, analyzing their connectivity, and exploring the potential role of urban heat islands could also serve as important research directions. These efforts may provide valuable insights into the establishment, survivability, and potential expansion of species that appear to exist outside of their bioclimatic optimum.

## Figures and Tables

**Figure 1 biology-14-00047-f001:**
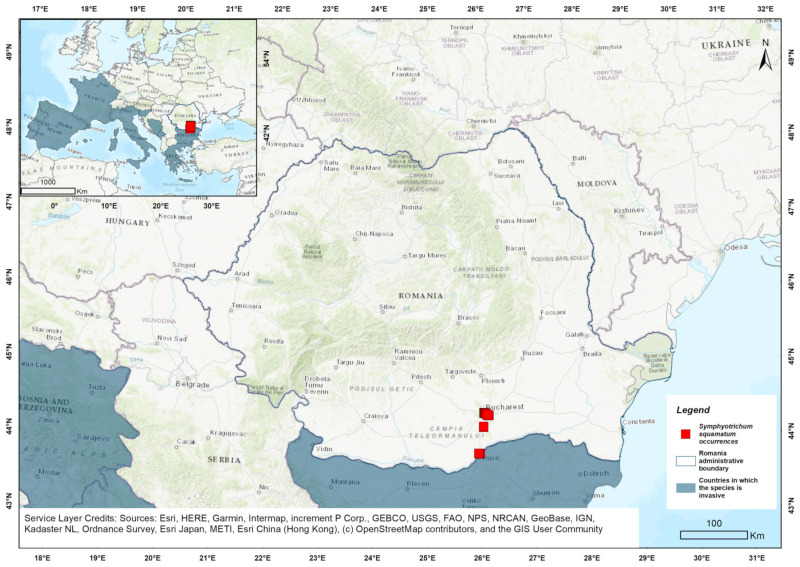
The distribution of *S. squamatum* in Romania, based on occurrence points collected from the field.

**Figure 2 biology-14-00047-f002:**
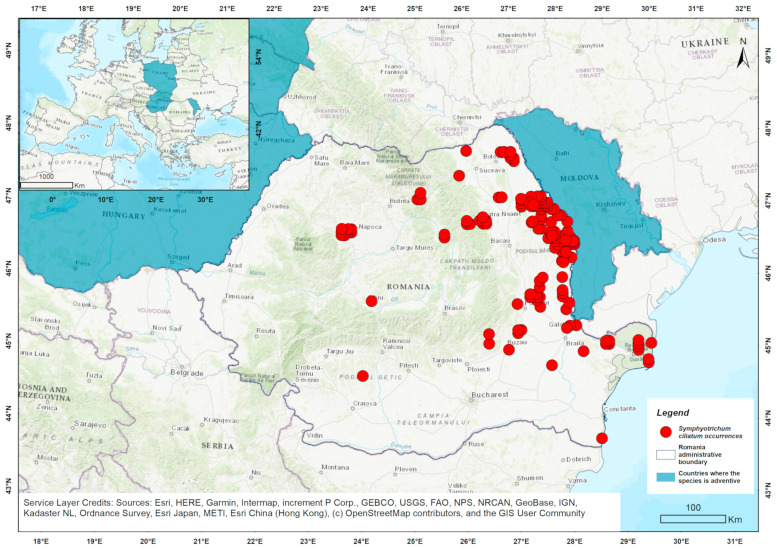
Distribution of *S. ciliatum* in Romania, based on occurrence data sourced from GBIF [69].

**Figure 3 biology-14-00047-f003:**
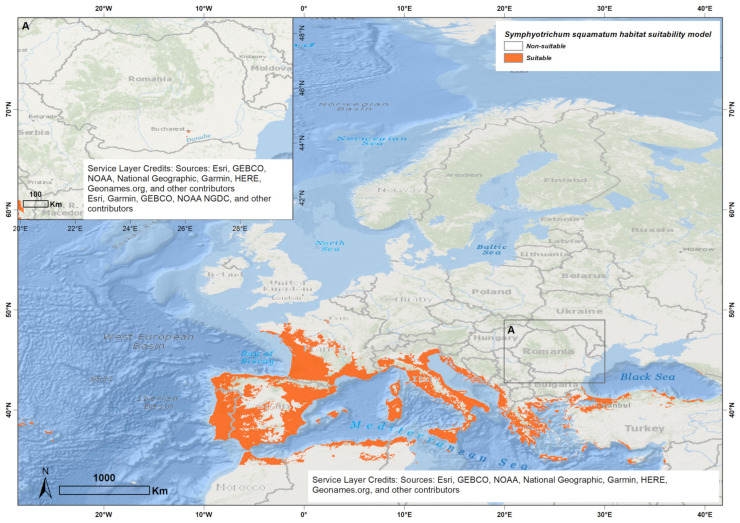
Potential habitat suitability of *S. squamatum* in Europe and in Romania.

**Figure 4 biology-14-00047-f004:**
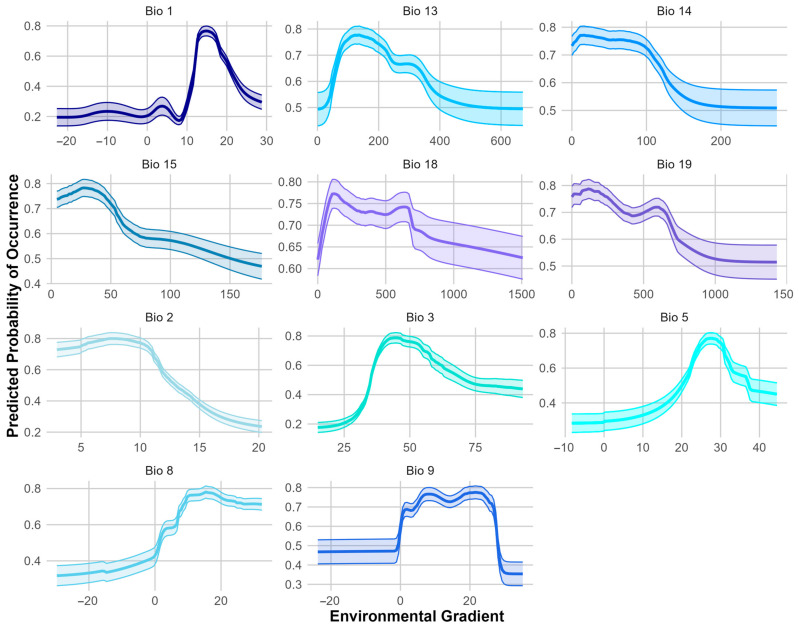
Response curves of the bioclimatic variables influencing the species distribution model (ESDM) for *S. squamatum*. The curves represent the predicted response of the species to the variation in each bioclimatic variable (Bio 1 = Annual Mean Temperature, Bio 13 = Precipitation of Wettest Month, Bio 14 = Precipitation of Driest Month, Bio 15 = Precipitation Seasonality, Bio 18 = Precipitation of Warmest Quarter, Bio 19 = Precipitation of Coldest Quarter, Bio 2 = Mean Diurnal Range, Bio 3 = Isothermality, Bio 5 = Max Temperature of Warmest Month, Bio 8 = Mean Temperature of Wettest Quarter, Bio 9 = Mean Temperature of Driest Quarter). The shaded areas around the curve indicate the 95% confidence intervals, representing the variability in the model predictions.

**Figure 5 biology-14-00047-f005:**
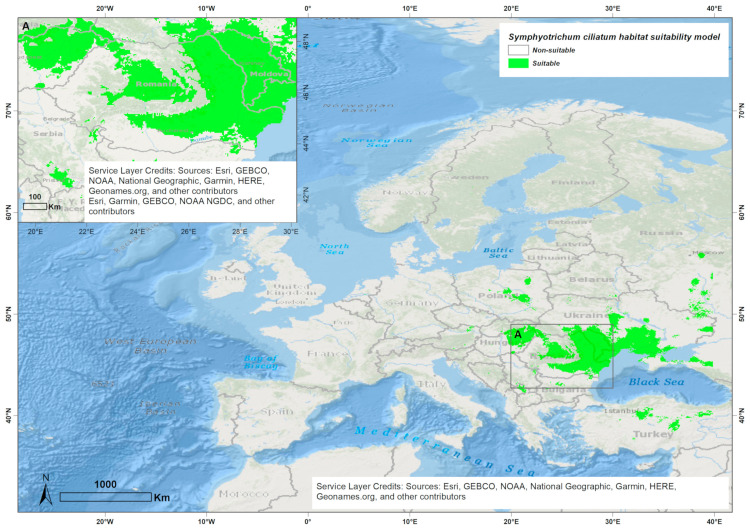
Potential habitat suitability of *S. ciliatum* in Europe and in Romania.

**Figure 6 biology-14-00047-f006:**
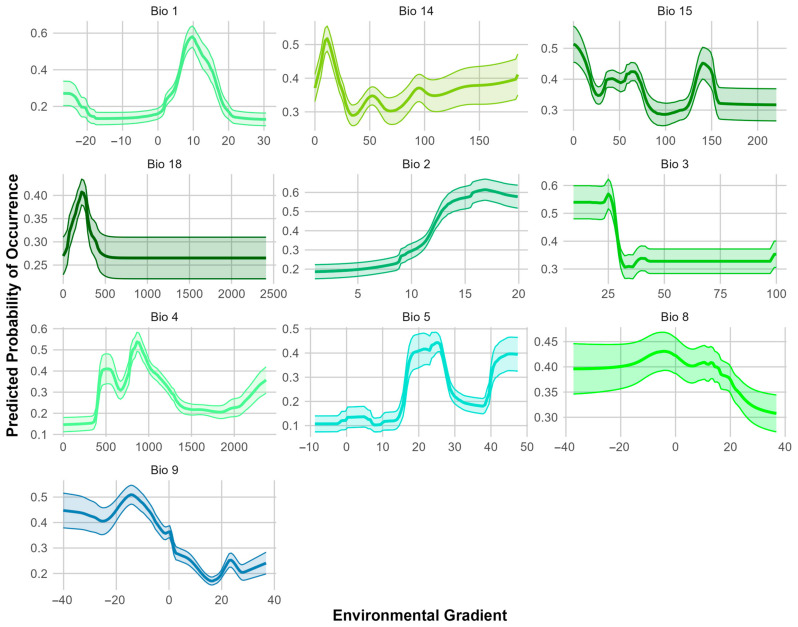
Response curves of the bioclimatic variables influencing the species distribution model (ESDM) for *S. ciliatum*. The curves represent the predicted response of the species to the variation in each bioclimatic variable (Bio 1 = Annual Mean Temperature, Bio 14 = Precipitation of Driest Month, Bio 15 = Precipitation Seasonality, Bio 18 = Precipitation of Warmest Quarter, Bio 2 = Mean Diurnal Range, Bio 3 = Isothermality, Bio 4 = Temperature Seasonality, Bio 5 = Max Temperature of Warmest Month, Bio 8 = Mean Temperature of Wettest Quarter, Bio 9 = Mean Temperature of Driest Quarter). The shaded areas around the curve indicate the 95% confidence intervals, representing the variability in the model predictions.

**Figure 7 biology-14-00047-f007:**
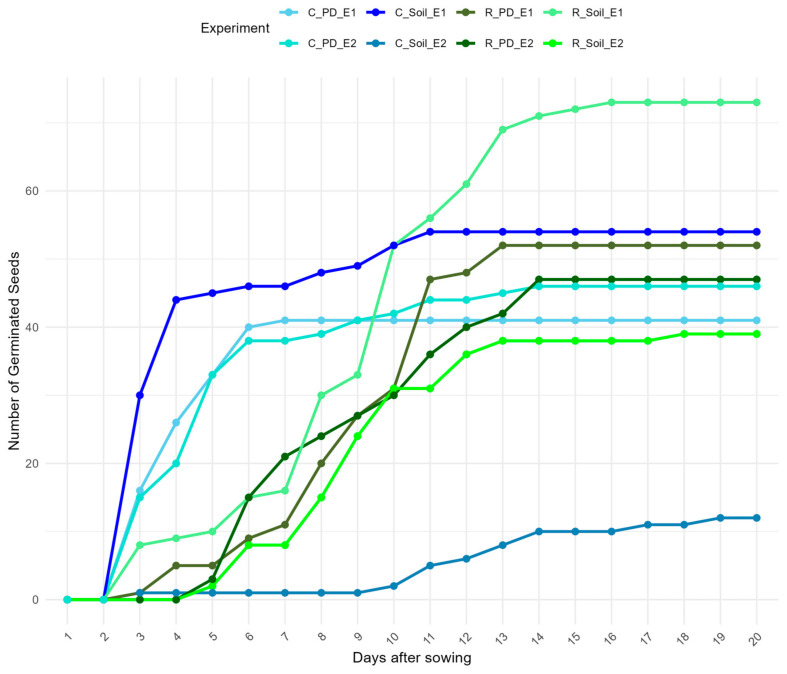
Germination dynamics of *Symphyotrichum squamatum* seeds collected from the Cotroceni (C_) and Rahova (R_) populations under different experimental conditions: E1—controlled temperatures; E2—room temperature; _PD_—seeds planted on Petri dish with moist filter paper; _Soil_—seeds planted in soil.

**Table 1 biology-14-00047-t001:** Performance metrics of the Ensemble Species Distribution Models (ESDMs) for *S. squamatum* and *S. ciliatum*.

Metric	*Symphyotrichum squamatum*	*Symphyotrichum ciliatum*
AUC (Area Under the Curve)	0.95	0.94
COR (Correlation)	0.7	0.65
Deviance	0.35	0.3
TSS (True Skill Statistic)	0.78	0.76
Sensitivity	0.91	0.89
Specificity	0.86	0.87

**Table 2 biology-14-00047-t002:** *S. squamatum*’s bioclimatic context range (considering the bioclimatic variables that contributed the most to the ESDM).

Variables	Description	Units	Romania (Mean ± SD)	Native Range (Mean ± SD)	European Invaded Range (Mean ± SD)
Bio 1	Annual Mean Temperature	°C	10.9 ± 0.4	17.8 ± 3	14.3 ± 1.9
Bio 3	Isothermality	%	31.9 ± 0.7	52.5 ± 8.6	38.7 ± 3.7
Bio 5	Max Temperature of Warmest Month	°C	29 ± 0.5	29.2 ± 3.4	28.5 ± 2.8
Bio 9	Mean Temperature of Driest Quarter	°C	1.4 ± 0.3	14.4 ± 3.4	19.9 ± 5.2
Bio 2	Mean Diurnal Range	°C	10.9 ± 0.1	11.6 ± 1.9	9.8 ± 1.6

**Table 3 biology-14-00047-t003:** *S. ciliatum*’s bioclimatic context range (considering the bioclimatic variables that contributed the most to the ESDM).

Variables	Description	Units	Romania (Mean ± SD)	Native Range (Mean ± SD)	European Invaded Range (Mean ± SD)
Bio 1	Annual Mean Temperature	°C	9.4 ± 1.3	5.4 ± 3.5	9.4 ± 1.2
Bio 5	Max Temperature of Warmest Month	°C	26.1 ± 1.5	26.7 ± 3.2	25.8 ± 1.7
Bio 4	Temperature Seasonality	°C × 100	855.8 ± 33.4	1104.3 ± 191.7	842.2 ± 66.4
Bio 2	Mean Diurnal Range	°C	9.5 ± 0.8	12.3 ± 2.4	9.4 ± 0.9
Bio 3	Isothermality	%	29.8 ± 1.9	29.2 ± 6.1	30 ± 2

**Table 4 biology-14-00047-t004:** Germination parameters for *S. squamatum* seeds from the Rahova and Cotroceni populations, under different substrate and temperature treatments.

Population	Substrate Type	Temperature Treatment	Germination Percentage (GP) (%)	Germination Energy (GE) (%)	Germination Rate Index (GRI) (%/day)
Rahova	Soil	Controlled Temperatures: 21 °C/8 °C (Experiment 1)	73	9	70.1
Rahova	Soil	Room Temperature: 23 °C (Experiment 2)	39	0	35.3
Rahova	Petri dishes with filter paper	Controlled Temperatures: 21 °C/8 °C (Experiment 1)	52	5	48.2
Rahova	Petri dishes with filter paper	Room Temperature: 23 °C (Experiment 2)	47	0	44.6
Cotroceni	Soil	Controlled Temperatures: 21 °C/8 °C (Experiment 1)	54	44	97
Cotroceni	Soil	Room Temperature: 23 °C (Experiment 2)	12	1	7.59
Cotroceni	Petri dishes with filter paper	Controlled Temperatures: 21 °C/8 °C (Experiment 1)	41	26	72.2
Cotroceni	Petri dishes with filter paper	Room Temperature: 23 °C (Experiment 2)	46	20	72.3

## Data Availability

Data are contained within the article and Appendix A.

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
