# Peer review of "Invasive Traits of Symphyotrichum squamatum and S. ciliatum: Insights from Distribution Modeling, Reproductive Success, and Morpho-Structural Analysis"

_biology, 2025, doi:10.3390/biology14010047_

Round 1
Reviewer 1 Report
Comments and Suggestions for Authors
I have made some suggestions for corrections and clarifications, with the hope that you will take them into consideration.
RECOMMANDATIONS
INTRODUCTION
Line 55: (13). Reference, the description found in the article could not be found in the reference.
Line 64: The abrevation (AIS) should be replace to (IAS).
How the soil coverage areas of invasive species were found and utilised.
How were the main topical characters identified?
How were field records collected?
What are the 19 bioclimatic variables? Should be explained.
In the conclusion and discussion; the objectives of the study, (1) the evaluation of bioclimatic conditions and habitat suitability for the two species, (2) the differences in the structural and reproductive characteristics of S. squamatum and S. ciliatum should be emphasised more clearly.
Reviewer 2 Report
Comments and Suggestions for Authors
Title: Invasive Traits of Symphyotrichum squamatum and S. ciliatum: Insights from Distribution Modeling, Reproductive Success, and Morpho-Structural Analysis
This research work address the question that is there any possibility of invasiveness of Symphyotrichum squamatum by studying comparative analysis with invasive traits of S. ciliatum, a well-established invasive species in Romania. This study established the fact using species distribution modelling, environmental variables (climatic, pedologic, human impact and topographical), reproductive traits assessment (germination percentage, germination energy and germination rate index) and comparative anatomical traits (preservation methods (70% alcohol), manual cross-sectioning (at standardized levels: median third of the primary root, lower and upper thirds of the main stem, and median third of the lamina), staining (using Iodine Green and Carmin Alum). Comprehensive account on already established invasive species with that of new invasive species were right choice for establishing the facts.
This study address the specific gap that the potential possible spread of new invasive species which is disrupt ecosystems, reduce biodiversity, and even pose risks to human health. Moreover, understanding invasive species habitat, reproduction etc. are essential for predicting and managing their ecological impacts.
This study is different from the published work as it established the fact S. squamatum, though this species has similar traits with successful invaders like S. ciliatum but Romania’s climate may limit its further spread.
This study used the appropriate methodology for characterizing factors responsible for invasive potential traits and factors. Detailed methodology adopted is given. Appropriate design and robust statistical analysis were followed.
Identifying a species’ potential to spread in the early stages of invasion is particularly important, as managing established invasive species is far more challenging, costly, and less effective.
Well discussed the results obtained in this study. Based on the extensive work done to prove the hypothesis, the conclusion drawn is appropriate.
Reviewer 3 Report
Comments and Suggestions for Authors
In the manuscript mention above authors have reported that while S. squamatum has similar traits with successful invaders, Romania’s climate may limit its further spread. This research may open new vistas to predict and manage plant invasions, which is essential for protecting ecosystems and reducing the impact of invasive spp.
The manuscript is well written
Introduction may include certain more recent references in the areas such as
characters associated with successful IAS establishment, including fitness homeostasis
extended flowering and fruiting periods and high propagule pressure.
iSDMS
RTA
In the Discussion section
Give more emphasis on
It is important to note that the potential invasiveness of S. squamatum does not depend on climatic factors .........
Reproductive success of S. squamatum
Add some more text in conclusion and future prospects and new directions in the research to be conducted in the area of the invasion success of alien species
Reviewer 4 Report
Comments and Suggestions for Authors
Thanks for presenting this multifaceted approach for assessing the invasion potential of alien species. The topic is of high interest for a broad audience and the paper is well written and easy to understand. The methods are almost always described in detail. Some critical issues concerning germination experiments and the comparative analysis of anatomical traits should be disentangled. Specific comments in the attached file.
